# Patterns of Maternal Neutrophil Gene Expression at 30 Weeks of Gestation, but Not DNA Methylation, Distinguish Mild from Severe Preeclampsia

**DOI:** 10.3390/ijms222312876

**Published:** 2021-11-28

**Authors:** Scott W. Walsh, Marwah Al Dulaimi, Kellie J. Archer, Jerome F. Strauss

**Affiliations:** 1Department of Obstetrics and Gynecology, Virginia Commonwealth University, Richmond, VA 23298-0034, USA; marwah.aldulaimi@vcuhealth.org (M.A.D.); jerome.strauss@vcuhealth.org (J.F.S.III); 2Division of Biostatistics, College of Public Health, The Ohio State University, Columbus, OH 43210, USA; archer.43@osu.edu

**Keywords:** preeclampsia, pregnancy, neutrophils, gene expression, DNA methylation, epigenetics, protease-activated receptor 1

## Abstract

Neutrophils are activated and extensively infiltrate blood vessels in preeclamptic women. To identify genes that contribute to neutrophil activation and infiltration, we analyzed the transcriptomes of circulating neutrophils from normal pregnant and preeclamptic women. Neutrophils were collected at 30 weeks’ gestation and RNA and DNA were isolated for RNA sequencing and 5-hydroxy-methylcytosine (5-hmC) sequencing as an index of dynamic changes in neutrophil DNA methylation. Women with normal pregnancy who went on to develop mild preeclampsia at term had the most uniquely expressed genes (697) with 325 gene ontology pathways upregulated, many related to neutrophil activation and function. Women with severe preeclampsia who delivered prematurely had few pathways up- or downregulated. Cluster analysis revealed that gene expression in women with severe preeclampsia was an inverse mirror image of gene expression in normal pregnancy, while gene expression in women who developed mild preeclampsia was remarkably different from both. DNA methylation marks, key regulators of gene expression, are removed by the action of ten-eleven translocation (TET) enzymes, which oxidize 5-methylcytosines (5mCs), resulting in locus-specific reversal of DNA methylation. DNA sequencing for 5-hmC revealed no differences among the three groups. Genome-wide DNA methylation revealed extremely low levels in circulating neutrophils suggesting they are de-methylated. Collectively, these data demonstrate that neutrophil gene expression profiles can distinguish different preeclampsia phenotypes, and in the case of mild preeclampsia, alterations in gene expression occur well before clinical symptoms emerge. These findings serve as a foundation for further evaluation of neutrophil transcriptomes as biomarkers of preeclampsia phenotypes. Changes in DNA methylation in circulating neutrophils do not appear to mediate differential patterns of gene expression in either mild or severe preeclampsia.

## 1. Introduction

Preeclampsia (PE) is a hypertensive disorder of human pregnancy that affects multiple maternal organs, placental function and, consequentially, fetal physiology. It occurs in 5–7% of all pregnancies, and is a leading cause of maternal and fetal morbidity and mortality [1]. Many pathologic alterations in cells and tissues have been associated with preeclampsia, but the cause of the disorder is still not known. Neutrophils may hold important keys to understanding the origins of the disease and the underlying causes of maternal organ dysfunction because they extensively infiltrate into maternal blood vessels in women with preeclampsia [2,3,4].

In 2012, we reported the first global assessment of DNA methylation in omental arteries of normal pregnant and preeclamptic women [5]. The most significant differences in DNA methylation were “hypomethylated” genes in preeclamptic women and gene pathway analysis revealed significant interconnectivity of hypomethylated genes with innate immune function. Many of these genes encoded neutrophil-related molecules and proteases, such as matrix metalloproteinase-1 (MMP-1), neutrophil elastase, TNFα, and thromboxane synthase [5]. We confirmed that reduced methylation of the MMP-1 and thromboxane synthase genes was associated with increased gene and protein expression in omental arteries of preeclamptic women [6,7]. The dominance of hypomethylated genes related to inflammation in blood vessels of preeclamptic women is quite remarkable and suggests that epigenetic alterations play an important role in the pathology.

It makes sense that the expression of inflammatory genes would be silenced in normal pregnancy. A mechanism for gene silencing could be DNA methylation, which masks binding sites for inflammatory transcription factors, such a NF-κB. However, if the methylation marks were erased, it would open these sites, possibly leading to increased gene expression. Enzymes capable of removing methylation marks have been discovered: TET proteins (ten-eleven translocation proteins, aka tet methylcytosine dioxygenases). These enzymes catalyze the conversion of 5-methycytosine (5-mC) to 5-hydroxy-methylcytosine (5-hmC) [8,9,10,11], which is further oxidized and then removed by the DNA base excision repair enzyme, thymine-DNA glycosylase, and replaced with unmodified cytosine [12]. Increases in DNA levels of 5-hmC indicate enzymatic removal of 5-mC, which is usually associated with increased gene expression. TET2 is the main TET protein expressed in leukocytes, and it is thought to play an essential role in regulating hematopoietic differentiation [13,14,15].

The present study was designed to test two hypotheses: (1) that neutrophils obtained from preeclamptic women are characterized by increased expression of genes related to neutrophil function and activation; and (2) that this differential gene expression is a consequence of alterations in DNA methylation mediated by TET2.

## 2. Results

### 2.1. RNA Sequencing of Pregnancy Neutrophils

Figure 1 shows a cluster analysis of differentially expressed genes comparing severe preeclampsia (SPE), mild preeclampsia (MPE) and normal pregnancy (NP). SPE gene expression was an inverse mirror image of NP. Genes downregulated in NP were upregulated in SPE, whereas genes upregulated in NP were downregulated in SPE. Gene expression for MPE was remarkably different from both SPE and NP. This is a significant finding because blood samples for MPE were collected when the women were still diagnosed with normal pregnancy. They did not develop clinical symptoms until approximately 8 weeks later.

Figure 2 shows a Venn diagram for transcriptomes for NP vs. MPE vs. SPE. There were 275 uniquely expressed genes in women with SPE, and 697 uniquely expressed genes in women who developed MPE later in pregnancy. In total, 8649 genes were expressed in neutrophils obtained from pregnant women.

The differentially expressed genes were subjected to gene ontology analysis. Women with normal pregnancy at 30 weeks’ gestation who went on to develop MPE at term had a large number of gene ontology pathways that were significant (325), and all were upregulated as compared to women with NP (Table 1, Figure 3). Many of these pathways were specifically related to neutrophil function and activation. For example, leukocyte differentiation, megakaryocyte differentiation, regulation of cell migration, multicellular response to stress, blood vessel endothelial migration, angiogenesis, tissue migration, coagulation, hemostasis, positive regulation of protein serine/threonine activity, activation of protein kinases, activation of MAP kinases, apoptosis, plasminogen activation, inflammatory response, hemopoiesis, leukocyte degranulation, nitric oxide signal transduction, endothelial cell chemotaxis, TNFα biosynthesis, TGF beta1 production and granulocyte differentiation. These data indicate that changes in neutrophil gene expression pathways occur very early in women who go on to develop MPE. In contrast, women with SPE who delivered preterm shortly after sample collection (range 1 day to 3 weeks) had only 16 pathways significantly upregulated and 15 significantly downregulated as compared to NP (Table 2, Figure 3), and few were related to neutrophil function. Two pathways related to leukocyte apoptosis were upregulated and six pathways related to neutrophil function and activation were downregulated.

### 2.2. hMeDIP (5-hmC) DNA Sequencing of Pregnancy Neutrophils

Over 30,000 genes were identified by DNA sequencing with peaks found within 10 kb of the gene start/end for 5-hmC in neutrophils obtained from pregnant women. Merged Regions analysis, which is the most useful for comparison of the samples, did not identify any significant differences among groups. This is graphically illustrated in Figure 4 and Figure 5. Figure 4 shows the normalized and raw log2 tag counts for NP, MPE and SPE for which there were no differences in 5-hmC marks and gene expression. Figure 5 shows a principal component analysis scatterplot for NP, MPE and SPE. There were no clusters that separated the groups.

### 2.3. DNA Methylation in Leukocytes

Figure 6 shows DNA methylation identified in the buffy coat leukocytes as compared to tissue levels in omental arteries [5] for normal pregnant and severe preeclamptic women. Total DNA methylation in leukocytes was extremely low in both NP and SPE (b-values < 0.1) and significantly lower than that present in omental arteries (b-values of about 0.4, *p* < 0.0001). The dataset for leukocytes was filtered to obtain probes interrogating promotor-associated CpG sites in CpG islands. After filtering, there were 5078 CpG sites for analysis. The smallest FDR was 0.3224. Using a less stringent filter (changed max (βij) − min (βij) > 0.17 to max (βij) − min (βij) > 0.10), we tested 13,259 CpG sites; however, the smallest FDR in this expanded set was 0.3823. Therefore, there was no significant difference between normal and preeclamptic pregnancy for total DNA methylation in circulating leukocytes. This may explain the lack of significant differences among groups for 5-hmC because this suggests circulating leukocytes are already de-methylated. Given the small number of MPE subjects in our study, and the fact that indicated surgical delivery in MPE is rare, we were unable to perform methylation studies on omental arteries in this subset of PE.

## 3. Discussion

Analysis of gene expression in neutrophils obtained from a multiracial/ethnic population of preeclamptic women revealed some interesting findings. Surprisingly, cluster analysis of differentially expressed genes revealed that gene expression in women with SPE was an inverse mirror image of normal pregnancy. Genes with low expression in normal pregnancy had high expression in SPE, whereas genes with high expression in normal pregnancy had low expression in SPE. Just as surprising was the fact that few gene ontology pathways were up- or downregulated in SPE. Downregulated gene pathways were related to neutrophil activation.

Another remarkable finding was that women who developed MPE later in their pregnancy had the most profound alterations in expressed neutrophil genes with 697 uniquely expressed. Furthermore, an astounding 325 gene ontology pathways were significant, and remarkably, all were upregulated, with many related to neutrophil activation and function. These samples were collected more than 8 weeks before clinical symptoms appeared while the women were still considered to have a normal pregnancy. This suggests that gene expression changes leading to MPE are occurring long before clinical symptoms manifest. Pertinent to this is the finding that levels of MMP-1 are significantly elevated in maternal plasma 8 weeks before the emergence of clinical symptoms of PE, and that MMP-1 activates pregnancy neutrophils via pregnancy-specific expression of the protease-activated receptor 1 (PAR-1) [16]. This implies that neutrophil gene expression in PE may be unique to pregnancy. This notion is supported by a recent paper identifying a unique genetic signature in circulating RNA transcripts in women with PE [17].

Neutrophils extensively infiltrate maternal blood vessels causing inflammation in women with SPE [2,3,4], so it was puzzling that neutrophil activation pathways in circulating neutrophils were downregulated in women with SPE. We speculate that this was a consequence of vascular infiltration. That is, new neutrophils were entering the circulation from the bone marrow to replace the neutrophils that had left. This would be necessary to maintain circulating concentrations and would be consistent with the activation pathways of these new neutrophils being downregulated. In contrast, neutrophil activation pathways in women who went on to develop MPE were upregulated, but apparently not to the point of causing vascular infiltration, so they remained in the circulation with upregulated pathways.

The striking differences in transcriptomes of maternal neutrophils suggest that MPE and SPE result from different underlying pathophysiological mechanisms of disease. This idea is supported by a recent comprehensive evaluation of placental lipids in women who developed preeclampsia on low-dose aspirin therapy [18]. Women who developed SPE preterm had significant elevations in non-aspirin-sensitive eicosanoids and sphingolipids with biological actions that could cause PE, but the lipid profile in women who developed MPE did not show these elevations and their lipid profile was no different than women with normal pregnancy.

There were some differences between women who developed MPE versus SPE, but we do not believe these can explain the differences in gene expression. Women with SPE were treated with magnesium sulfate and anti-hypertensive drugs, but we are not aware of any evidence that these drugs can affect gene expression. The presence of risk factors or aspirin cannot explain the differences because only four of the women with SPE had pre-existing risk factors and only two had taken aspirin. Perhaps more thought-provoking was the dramatic difference in gene expression between women who developed MPE versus women who had normal pregnancies. There was no apparent difference between these groups at 30 weeks’ gestation. They were both diagnosed with normal pregnancy, neither group was prescribed aspirin or other drugs, and neither group had risk factors for PE. The only difference was the dramatic difference in gene expression.

Another surprising finding was that there was no difference in 5-hmC methylation levels between normal and preeclamptic pregnancies. However, upon further reflection, this makes sense. Circulating neutrophils are mature cells that do not divide and have a limited gene expression profile [19,20]. Differentiation of neutrophils occurs in the bone marrow, so TET2 de-methylation would have had to occur in the bone marrow to open transcription factor binding sites for differentiation. Thus, the 5-hmC data do not invalidate our hypothesis regarding TET2 regulation of gene expression, it only indicates that circulating neutrophils are already “de-methylated”. Consistent with this notion, less than 10% of the DNA in circulating leukocytes was methylated as compared to about 40% for omental arteries. With over 30,000 genes identified by 5-hmC analysis, but only about 8600 by RNA sequencing, it appears that TET2 de-methylation in neutrophils is not locus-specific, but rather affects the entire genome.

Our results, which suggest that the entire genome of circulating neutrophils is de-methylated, are supported by the additional finding that many of the gene ontogeny pathways upregulated in women who went on to develop MPE had nothing to do with neutrophil function. For example, pathways for endothelial cells, epithelial cells, neurons, fibroblasts and Sertoli cells among others were significantly upregulated. The physiologic benefit associated with the upregulation of these pathways is not known, but it is consistent with the notion that the genome of circulating neutrophils is de-methylated. If the entire genome is indeed de-methylated as suggested by the 5-hmC results identifying over 30,000 genes, genes unrelated to neutrophil function could be transcribed by activating mechanisms, such as PAR-1-induced phosphorylation to activate transcription factors that regulate genes not related to neutrophil function [16]. PAR-1 is constitutively expressed in many other cell types, including other leukocytes, platelets, endothelial cells, smooth muscle cells, and neurons [21], so protease activation of neutrophil PAR-1 may activate these other pathways even though they have nothing to do with neutrophil function.

Although a limitation of our studies includes the small number of subjects studied and sampling at a single time point, the dramatic differences detected in gene expression provide a strong rationale for exploring neutrophil transcriptome profiles as biomarkers for predicting different PE phenotypes. Future studies should include larger cohorts and sampling at multiple time points, particularly earlier in gestation.

## 4. Materials and Methods

### 4.1. Study Subjects

Gestational-age-matched blood samples were collected at approximately 30 weeks’ gestation from a multi-racial/ethnic population of women with normal pregnancy who delivered at term (*n* = 11, NP), women with severe preeclampsia who delivered preterm at <37 weeks (*n* = 10, SPE) and from women with normal pregnancy at 30 weeks who later developed mild preeclampsia at term (*n* = 3, MPE). These samples were used for RNA and DNA sequencing after isolating the neutrophils. In addition, blood samples were collected from a separate group of women with NP (*n* = 4) or SPE (*n* = 5) for collection of the buffy coat for global DNA methylation analysis. Normal pregnant women had maternal blood pressures ≤110/70 mmHg, no proteinuria, and no other complications. Preeclamptic women had blood pressures of ≥140/90 mmHg on 2 occasions at least 4 h apart after 20 weeks’ gestation and proteinuria (protein/creatinine ratio ≥ 0.3). Medications used to treat women with severe preeclampsia included magnesium sulfate, labetalol, and hydralazine. According to the medical records, only two of the women with severe preeclampsia had taken aspirin. Only four were recruited from our high-risk clinic with pre-existing risk factors (chronic hypertension, pre-gestational diabetes). None of the women with SPE were in labor. All vaginal deliveries were induced. Women with normal pregnancy at 30 weeks who went on to develop MPE at term were not prescribed aspirin or other drugs and had no risk factors for PE. The Office of Research Subjects Protection of Virginia Commonwealth University approved this study (HM20009145). All subjects gave informed consent, and the procedures followed were in accordance with institutional guidelines. Clinical characteristics of the patient groups are given in Table 3.

### 4.2. RNA Sequencing and hMeDIP (5-hmC) DNA Sequencing of Neutrophils

Two 10 mL heparin tubes of blood were collected from pregnant women. Lymphocytes and monocytes were separated from granulocytes (96% of which are neutrophils) by Histopaque (1077/1119) density gradient centrifugation according to the manufacturer’s protocol (Sigma Aldrich, St. Louis, MO, USA) and as previously described [3,22,23]. DNA and RNA were isolated from neutrophils using TRIzol Reagent (ThermoFisher Scientific, Wilmington, DE, USA) according to the manufacturer’s protocol. RNA and DNA concentrations were measured, and their quality assessed using a NanoDrop 2000 spectrophotometer (ThermoFisher Scientific). Total RNA (0.5 µg/20 µL) was sent to Novogene Corporation, Inc. (Sacramento, CA, USA) for Human mRNA Sequencing. RNA sample quality was determined by Novogene before proceeding with mRNA library preparation (poly A enrichment). A paired-end 150 bp sequencing strategy was used to sequence the samples using an Illumina NovaSeq 6000 Sequencing Platform. The resulting data were checked for their quality before bioinformatic analyses. The hg38 genome was used as the reference genome for gene alignment. Novogene provided the bioinformatics analysis for the RNA-seq Quantification Analysis Report along with publication-ready results. Intact DNA (20 µg at concentrations > 390 ng/µL) was sent to Active Motif (Carlsbad, CA, USA) for MethylPath hMeDIP-sequencing to assess the activity of the TET2 enzyme. DNA was sonicated to 150–300 bp and Illumina adaptors were ligated to the DNA ends. This DNA was then used in IP (immuno-precipitation) reactions using antibody AM39791 (anti 5-hmC). IP-ed DNA and Input DNA that did not go thought the IP step were processed into sequencing libraries and sequenced using the Illumina platform (NextSeq 500, 75-nt single-end). The hg38 genome was used for gene alignment. Hydroxymethylated regions were identified using the MACS peak finding algorithm. Active Motif provided the bioinformatics analysis along with the explanation of data, differential analyses and graphs.

### 4.3. DNA Methylation in Leukocytes

Two 10 mL heparin tubes of blood were collected, pooled and then centrifuged (1000× *g*) to separate plasma from the buffy coat and red blood cells. The buffy coat layered on top of the red blood cells was aspirated for isolation of leukocyte DNA using a QuickGene DNA tissue kit S kit and QuickGene-Mini80 system (AutoGen, Holliston, MA, USA) according to the manufacturer’s protocol. RNase treatment was performed with RNase A (Qiagen, Valencia, CA, USA). DNA concentration was measured, and its quality was assessed using a NanoDrop 2000 spectrophotometer. DNA at 50 ng/µL totaling 1µg was sent to HudsonAlpha Clinical Services Lab (Huntsville, AL, USA). DNA was bisulfite-treated prior to genome-wide DNA methylation analysis using the Illumina Infinium HumanMethylation 450K Bead Chip assay (Illumina, San Diego, CA, USA). Scanned arrays were processed using Illumina’s GenomeStudio Methylation Analysis Module to obtain the β values for each probe, where βij represents proportion methylated for the ith probe and the jth array. The dataset was then filtered to obtain probes interrogating promotor-associated CpG sites in CpG islands. Data analysis was performed as previously described [5] using the beadarray package in an R programming environment [24]. To control for multiple hypothesis testing, the *p*-values were subsequently used in estimating the false discovery rates (FDRs) using the *q*-value method [25]. Methylation values (β values) are expressed as range from 0 to 1 where 0 means not methylated and 1 means fully methylated.

## 5. Conclusions

Our findings demonstrate that neutrophil gene expression profiles can distinguish different PE phenotypes (e.g., mild vs. severe). In MPE, alterations in gene expression can occur well before clinical symptoms appear, raising the possibility that neutrophil transcriptome profiling can serve as a biomarker of impending PE.

Changes in DNA methylation in circulating neutrophils do not appear to mediate the differential patterns of gene expression. The gene expression alterations are correlated with elevations in circulating proteases, such as MMP-1, that activate neutrophils based on the pregnancy-specific expression of PAR-1. The most plausible explanation for the extraordinary low levels of global DNA methylation in circulating neutrophils, as well as other leukocytes, is that they are “de-methylated” during differentiation in the bone marrow.

## Figures and Tables

**Figure 1 ijms-22-12876-f001:**
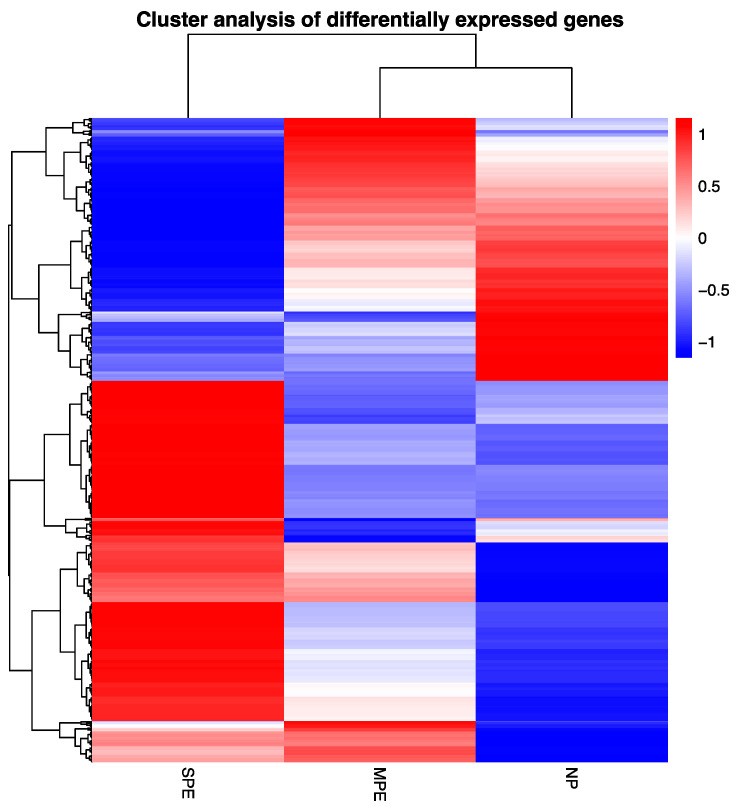
Cluster analysis of differentially expressed genes comparing severe preeclampsia (SPE), mild preeclampsia (MPE) and normal pregnancy (NP). SPE gene expression (*n* = 10) was an inverse mirror image of NP (*n* = 11), whereas MPE (*n* = 3) was different than both SPE and NP. Red indicates upregulated genes and blue indicates downregulated genes.

**Figure 2 ijms-22-12876-f002:**
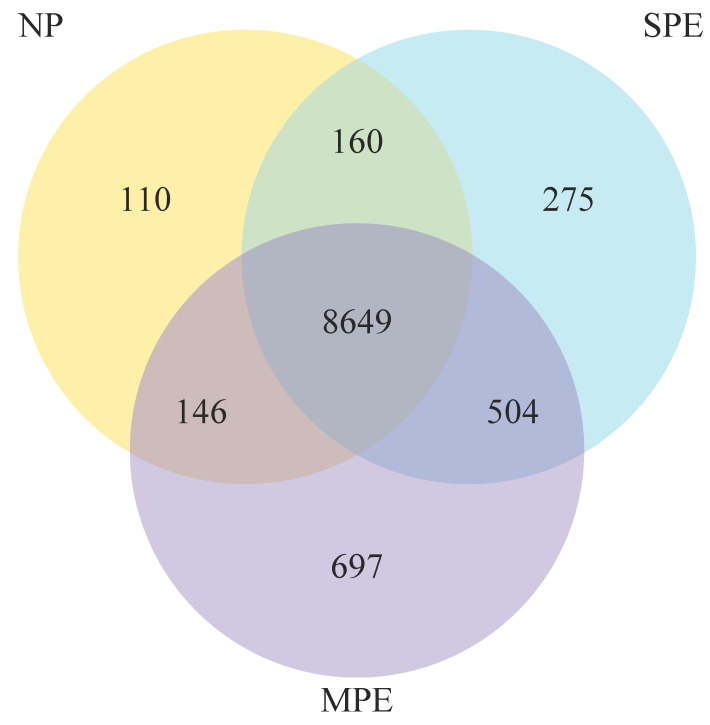
Venn diagram for RNA sequencing for NP (*n* = 11) vs. MPE (*n* = 3) vs. SPE (*n* = 10). In total there were 8649 genes expressed in pregnancy neutrophils, with 110 uniquely expressed in NP, 697 uniquely expressed in MPE and 275 uniquely expressed in SPE.

**Figure 3 ijms-22-12876-f003:**
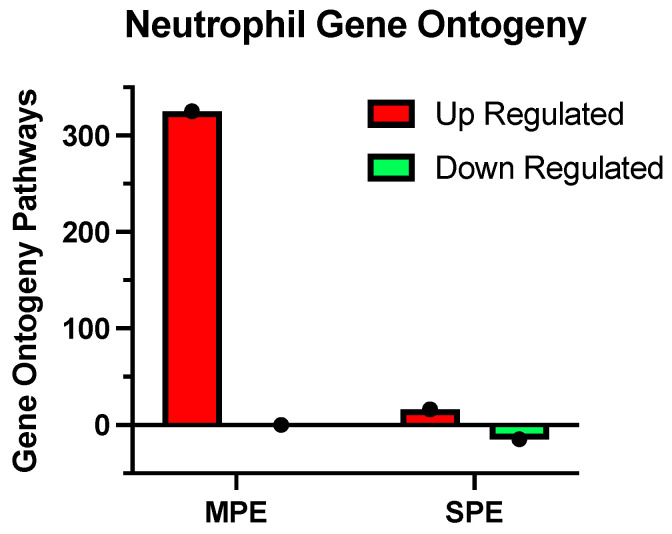
Comparison of significantly expressed gene ontogeny pathways in women diagnosed with normal pregnancy at approximately 30 weeks’ gestation who went on to develop MPE 8 weeks later with those of women with SPE who delivered prematurely shortly after sample collection.

**Figure 4 ijms-22-12876-f004:**
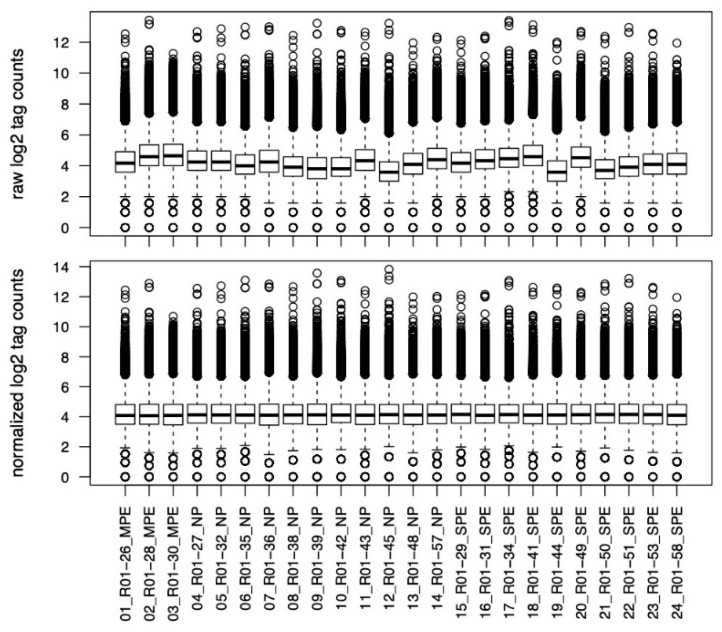
Normalized and raw log2 tag counts for 5-hmC sites identified in neutrophils obtained from women with NP (*n* = 11), MPE (*n* = 3) and SPE (*n* = 10). No significant differences were present among groups.

**Figure 5 ijms-22-12876-f005:**
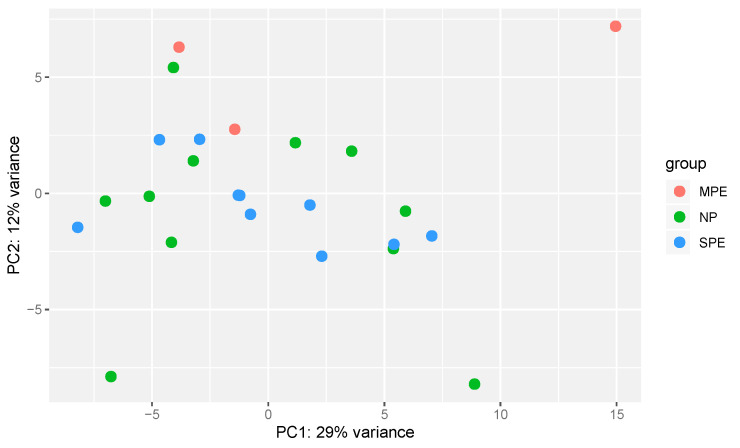
Principal component analysis in neutrophils of women with NP, MPE and SPE. There was no clustering for any group. No significant differences were present.

**Figure 6 ijms-22-12876-f006:**
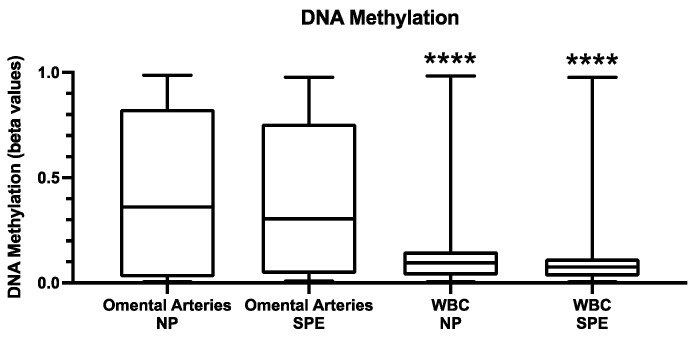
DNA methylation identified in the buffy coat leukocytes as compared to that identified in omental arteries of women with NP (*n* = 4) and women with SPE (*n* = 5). Box-and-whisker plots with error bars representing minimum to maximum. **** *p* < 0.0001 as compared to omental arteries.

**Table 1 ijms-22-12876-t001:** Selected neutrophil gene ontogeny pathways that were upregulated at 30 weeks’ gestation in women who went on to develop mild preeclampsia at term as compared to women at 30 weeks’ gestation who went on to have a normal pregnancy at term.

• Regulation of blood vessel endothelial cell proliferation involved in sprouting angiogenesis
• Blood vessel endothelial cell proliferation involved in sprouting angiogenesis
• Regulation of endothelial cell apoptotic process
• Endothelial cell apoptotic process
• Regulation of cell migration involved in sprouting angiogenesis
• Multicellular organismal response to stress
• Cell migration involved in sprouting angiogenesis
• Positive regulation of blood vessel endothelial cell migration
• Regulation of megakaryocyte differentiation
• Megakaryocyte differentiation
• Phagocytosis, engulfment
• Positive regulation of endothelial cell migration
• Regulation of blood vessel endothelial cell migration
• Sprouting angiogenesis
• Blood vessel endothelial cell migration
• Regulation of endothelial cell proliferation
• Endothelial cell proliferation
• Regulation of endothelial cell migration
• Positive regulation of angiogenesis
• Positive regulation of vasculature development
• Endothelial cell migration
• Regulation of myeloid cell differentiation
• Positive regulation of MAP kinase activity
• Tissue migration
• Regulation of angiogenesis
• Blood coagulation
• Hemostasis
• Coagulation
• Activation of protein kinase activity
• Positive regulation of protein serine/threonine kinase activity
• Regulation of MAP kinase activity
• Regulation of cGMP-mediated signaling
• Regulation of transforming growth factor beta1 production
• Regulation of vasculature development
• Phagocytosis
• Transforming growth factor beta1 production
• Regulation of fibrinolysis
• Positive regulation of leukocyte activation
• Programmed cell death involved in cell development
• Positive regulation of fibroblast migration
• Epithelial cell proliferation
• Positive regulation of cell activation
• Chemosensory behavior
• Regulation of plasminogen activation
• Positive regulation of phagocytosis, engulfment
• Myeloid cell differentiation
• Regulation of phagocytosis, engulfment
• Fatty acid transmembrane transport
• Regulation of endothelial cell chemotaxis
• Chronic inflammatory response
• Plasminogen activation
• Regulation of hemopoiesis
• Positive regulation of tumor necrosis factor biosynthetic process
• Positive regulation of MAPK cascade
• Regulation of protein serine/threonine kinase activity
• Positive regulation of blood coagulation
• Regulation of macrophage differentiation
• Positive regulation of hemostasis
• Positive regulation of cell migration
• Leukocyte differentiation
• Angiogenesis
• Positive regulation of protein kinase activity
• Nitric-oxide-mediated signal transduction
• Positive regulation of leukocyte degranulation
• Positive regulation of coagulation
• cGMP-mediated signaling
• Positive regulation of transforming growth factor beta receptor signaling pathway
• Positive regulation of cellular response to transforming growth factor beta stimulus
• Regulation of fatty acid transport
• Positive regulation of macrophage activation
• Tumor necrosis factor biosynthetic process
• Regulation of tumor necrosis factor biosynthetic process
• Granulocyte differentiation
• Regulation of granulocyte chemotaxis

Differentially expressed genes identified by RNA-seq were subjected to gene ontology analysis by Novogene. The adjusted *p*-value (padj), which is the transformation of the *p*-value after accounting for multiple testing, was used to determine statistical significance for these pathways. The padj value was calculated by Novogene and ranged from 0.003 to 0.03.

**Table 2 ijms-22-12876-t002:** Selected neutrophil gene ontogeny pathways that were upregulated or downregulated in women at 30 weeks’ gestation who had severe preeclampsia as compared to women at 30 weeks’ gestation who had normal pregnancies.

A. Upregulated Pathways
• Regulation of leukocyte apoptotic process
• Leukocyte apoptotic process
The adjusted *p*-value (padj) for these pathways was < 0.05
B. Downregulated Pathways
• Neutrophil degranulation
• Neutrophil activation involved in immune response
• Neutrophil-mediated immunity
• Neutrophil activation
• Granulocyte activation
• Tumor-necrosis-factor-mediated signaling pathway
Differentially expressed genes identified by RNA-seq were subjected to gene ontology analysis by Novogene. The adjusted *p*-value (padj) for these pathways was < 0.0001.

**Table 3 ijms-22-12876-t003:** Clinical characteristics of patient groups for RNA and 5-hmC sequencing.

Variable	NP *n* = 11	MPE *n* = 3	SPE *n* = 10
Maternal age (years)	29.8 ± 5.6	31.3 ± 7.0	28.4 ± 2.2
Pre-pregnancy BMI (kg/m^2^)	27.6 ± 5.8	27.5 ± 5.5	32.8 ± 10.0
BMI at sample collection (kg/m^2^)	32.2 ± 6.1	33.7 ± 7.6	38.0 ± 10.6
Systolic blood pressure at 30 weeks (mmHg)	112 ± 9	116 ± 12	179 ± 16 ****
Diastolic blood pressure at 30 weeks (mmHg)	69 ± 8	77 ± 13	109 ± 12 ****
Protein/creatinine ratio	ND	0.4 ± 0.1	1.2 ± 1.4
Primiparous	5	1	6
Multiparous	6	2	4
Race			
White	6	2	3
Black	2		4
Hispanic	1	1	3
Asian	2		
Type of delivery C-section Vaginal	2 9	2 1	4 6
Gestational age at sample collection (weeks)	32.6 ± 2.3	30.5 ± 2.1	31.8 ± 4.2
Gestational age at delivery (weeks)	38.6 ± 1.3	38.9 ± 3.2	32.8 ± 4.2 ***
Infant birth weight (grams)	2986 ± 476	2818 ± 836	1700 ± 670 ***

Values are mean ± SD. *** *p* < 0.001, **** *p* < 0.0001 as compared to NP and MPE. ND = not determined.

## Data Availability

Complete tabular data presented in this study for gene ontogeny pathways and gene expression are available on request from the corresponding author.

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
