# Peer review of "Patterns of Maternal Neutrophil Gene Expression at 30 Weeks of Gestation, but Not DNA Methylation, Distinguish Mild from Severe Preeclampsia"

_ijms, 2021, doi:10.3390/ijms222312876_

Round 1
Reviewer 1 Report
This is a study on gene expression regulation in neutrophils and DNA methylation in leukocytes in different types of preeclampsia (mild and severe) and healthy pregnant women in third trimester (30 weeks) of pregnancy.
Very interesting findings and conclusions are limited by small number of mild PE cases (3) which should be mentioned in manuscript and confirmed in wider population.
It would be very valuable to apply the study protocol earlier in pregnancy i.e. at 12 and 20 weeks to check if the results are similar.
It is not known what is the DNA methylation in the mild PE group (fig.6) which could enhance further understanding the difference between PE types.
Author Response
This is a study on gene expression regulation in neutrophils and DNA methylation in leukocytes in different types of preeclampsia (mild and severe) and healthy pregnant women in third trimester (30 weeks) of pregnancy.
Very interesting findings and conclusions are limited by small number of mild PE cases (3) which should be mentioned in manuscript and confirmed in wider population.
Thank you for noting this limitation of our study. We acknowledge this in the revised manuscript, but also note that the dramatic pathway differences between neutrophil transcriptomes detected in MPE and SPE form a strong basis for more detailed studies with larger numbers of subjects. On lines 239 – 243 we state:
Although a limitation of our studies includes the small number of subjects studied and sampling at a single time point, the dramatic differences detected in gene expression, provide a strong rationale for exploring neutrophil transcriptome profiles as biomarkers for predicting different PE phenotypes. Future studies should include larger cohorts and sampling at multiple time points, particularly earlier in gestation.
It would be very valuable to apply the study protocol earlier in pregnancy i.e., at 12 and 20 weeks to check if the results are similar.
This is an important point and we have noted this as a limitation of our study (see above)
It is not known what is the DNA methylation in the mild PE group (fig.6) which could enhance further understanding the difference between PE types.
Thank you for raising this point. We have addressed it by adding the following sentence to the Results section, Lines 153 - 155.
Given the small number of MPE subjects in our study, and the fact that indicated surgical delivery in MPE is rare, we were unable to perform methylation studies on omental arteries in this subset of PE.
Reviewer 2 Report
This is an interesting study which is well designed and developed for fulfilling the needs for predictive markers of preeclampsia. My only concern is that the authors didn't give an assessment of the possibilities of their proposal for such objective. I think the article must end with a consideration on the usefulness of the technique and markers in practice
Author Response
This is an interesting study which is well designed and developed for fulfilling the needs for predictive markers of preeclampsia. My only concern is that the authors didn't give an assessment of the possibilities of their proposal for such objective. I think the article must end with a consideration on the usefulness of the technique and markers in practice.
Thank you for this recommendation. We have revised the Abstract and the Discussion to point out that our studies provide potential biomarkers that distinguish the different PE phenotypes and that our observations offer a strong rationale for further studies evaluating the potential of neutrophil transcriptome characterization to predict the different PE phenotypes.
Abstract Lines 29-30:
These findings serve as a foundation for further evaluation of neutrophil transcriptomes as biomarkers of preeclampsia phenotypes.
Discussion, Lines 239-243:
Although a limitation of our studies includes the small number of subjects studied and sampling at a single time point, the dramatic differences detected in gene expression, provide a strong rationale for exploring neutrophil transcriptome profiles as biomarkers for predicting different PE phenotypes. Future studies should include larger cohorts and sampling at multiple time points, particularly earlier in gestation.
